# Increased visual and cognitive demands emphasize the importance of meeting visual needs at all distances while driving

**Amigale Patoine**[1]*, **Laura Mikula**[1], **Sergio Mejía-Romero**[1], **Jesse Michaels**[1], **Océane Keruzoré**[1], **Romain Chaumillon**[1], **Delphine Bernardin**[1,2], **Jocelyn Faubert**[1]

**1** Faubert Laboratory, School of Optometry, Université de Montréal, Montréal, Québec, Canada, **2** Essilor International Research and Development, Essilor Canada Ltd, Montréal, Québec, Canada

\* amigale.patoine@umontreal.ca

**Data Availability Statement:** The data underlying the results presented in the study are available from the Open Science Framework repository (osf. io/pmxuv/).

## Abstract

Having an optimal quality of vision as well as adequate cognitive capacities is known to be essential for driving safety. However, the interaction between vision and cognitive mechanisms while driving remains unclear. We hypothesized that, in a context of high cognitive load, reduced visual acuity would have a negative impact on driving behavior, even when the acuity corresponds to the legal threshold for obtaining a driving license in Canada, and that the impact observed on driving performance would be greater with the increase in the threshold of degradation of visual acuity. In order to investigate this relationship, we examined driving behavior in a driving simulator under optimal and reduced vision conditions through two scenarios involving different levels of cognitive demand. These were: 1. a simple rural driving scenario with some pre-programmed events and 2. a highway driving scenario accompanied by a concurrent task involving the use of a navigation device. Two groups of visual quality degradation (lower/ higher) were evaluated according to their driving behavior. The results support the hypothesis: A dual task effect was indeed observed provoking less stable driving behavior, but in addition to this, by statistically controlling the impact of cognitive load, the effect of visual load emerged in this dual task context. These results support the idea that visual quality degradation impacts driving behavior when combined with a high mental workload driving environment while specifying that this impact is not present in the context of low cognitive load driving condition.

## Introduction

With more than 1 billion motor vehicles in operation in the world as of 2010 and an estimated increase to 2 billion for 2030 [1], driving is one of the most dominant types of transportation. Worldwide, the overall number of road traffic deaths reached 1.35 million in 2016 and road traffic injury has been identified as the 8th leading cause of death for people of all ages [2]. Vision is undoubtedly important in driving safety as it allows drivers to perceive the road clearly and anticipate unexpected dangerous events [3]. Regardless of individual countries'

**Funding:** This work was supported by the Natural Sciences and Engineering Research Council of Canada, NSERC – Essilor Industrial Research Chair in the form of a grant awarded to AP and LM (IRCPJ 305729-13), Research and development cooperatif NSERC in the form of an Essilor Grant awarded to JF and DB (CRDPJ 533187 - 2018), Prompt (https://www.nserc-crsng.gc.ca/index_eng.asp & https://www.essilor.ca) in the form of funds awarded to DB, the Road Safety Research Network (Réseau de Recherche en Sécurité Routière) of Québec (https://rrsr.ca/en) in the form of student awards to LM and AP, CooperVision in the form of materials used for the study, and Essilor Canada Ltd. in the form of a salary for DB. The specific roles of these authors are articulated in the 'author contributions' section. The funders had no role in study design, data collection and analysis, decision to publish, or preparation of the manuscript.

**Competing interests:** The authors have read the journal's policy and have the following competing interests: DB is an adjunct professor of the NSERC/Essilor Chair, RDC NSERC-Essilor project at Université de Montréal and is employed by Essilor Canada as research project manager. Essilor also provided research grant support for this study. CooperVision provided the contact lenses used for the degradation of visual quality of participants. This does not alter our adherence to PLOS ONE policies on sharing data and materials. There are no patents, products in development or marketed products associated with this research to declare.

regulations, all drivers must meet a minimum visual acuity requirement in order to obtain and, possibly, renew their driving license [4]. For non-commercial drivers in Québec, Canada (Class 5 driver's license), the visual acuity should be at least 20/50 as measured by a Snellen chart with or without corrective lenses, both eyes examined together [5]. While there are striking disparities across countries' driving visual standards [4, 6], the International Council of Ophthalmology has recommended a minimum visual acuity of 20/40 for drivers [7].

Even though static visual acuity is the most common clinical test taken into consideration when applying for the driver's license, there is a clear lack of scientific justification to support current visual acuity standards [6, 8–13]. It has been reported that simulated visual acuity alteration mainly affects road sign detection and road hazard management behavior [14, 15] when reduced below 20/40, whereas the ability to operate a vehicle is not significantly impaired until vision is degraded to 20/100 or below [14, 16]. As another example, steering accuracy and lane keeping have been shown to remain relatively stable, even in the presence of high levels of blur going up to 6/197 [17, 18]. In fact, the relevant literature fails to reveal direct evidence for the role of visual acuity in accidentology and risky driving behavior [14, 19], thus suggesting that static visual acuity measurement alone might not be a reliable predictor of driving safety [3, 20]. Indeed, Wood et al. highlighted that increased cognitive demands associated with experimentally induced visual perturbations (blur or cataract) interfered with driving behavior [21]. More specifically, they reported longer time to complete a course, more hazard hits as well as impaired road sign recognition in presence of a secondary task on the dashboard, and these effects were more pronounced in impaired than in normal vision conditions. This result emphasizes the importance of both visual and cognitive demands to better account for the effects of visual perturbations during driving.

In accordance with this statement, other studies have focused on the interaction between vision and cognition while driving by investigating the role of visuo-cognitive capacities in road safety. A large number of studies have studied the Useful Field Of View (UFOV) in order to shed light on the link between driving behavior and perceptual-cognitive capacities. This perceptual-cognitive test is divided into three subtests which assess visual processing speed (UFOV 1), divided attention (UFOV 2) and selective attention (UFOV 3) capacities. The obtention of low scores on the UFOV 2 and 3 subtests have been shown to be predictive of road accidents [22–24] and driving behavior [24]. More recently, the score obtained in the 3-dimensional multiple object tracking (3D-MOT) assessing visual processing speed, working memory as well as selective, divided, distributed and sustained attention [25] has been demonstrated as a relevant predictor of driving behavior in a driving simulator [26, 27]. Emphasizing the importance of visuo-cognitive capacities in daily dynamic activities, Michaels et al. also indicates that task workload is a relevant indicator for revealing differences in driving behavior between individuals [26]. This evidence suggests that higher-order visual processing, such as visual attention, is an important component in driving behavior [28, 29] alongside good visual sensory capacities.

Since the beginning of the 20th century, the evolution of car design has not only improved driver comfort and safety, but has also modified the visual content and display of the information in the dashboard. This modification is believed to have led to an increase in the visual and cognitive resources needed while driving [30]. With the arrival of new technologies such as smartphones, heads-up displays and multiple options of tangible user interfaces to display navigation or vehicle information, the visual demand on the car dashboard, which in current modern times also includes all the interior front devices—and thus intermediate distance demands—has considerably increased. In addition to the far distance visual acuity, the closer visual acuity demand becomes more prominent, which requires additional binocular accommodation and convergence capacities. Moreover, frequent reallocation of attentional and

visual focus between driving tasks at multiple distances represents a challenge for finite mental resources [31]. The use of in-vehicle devices requires drivers' to use multitasking strategies in order to preserve safe interaction between visual and cognitive mechanisms. For example, higher glance frequency at road intersections has been indicated to result in increased visuo-cognitive engagement and therefore provides greater road safety in the elderly [32]. This suggests that failure to meet multiple visual demands can result in a potential risk of distraction because of limited attentional resources available to switch between the road and the car dashboard. Therefore, risk will potentially also exist in presence of uncorrected myopic defocus, refractive blur or decreased binocular integration. In fact, these were demonstrated to affect the response time to read [33], the cueing detection in a contextual cueing paradigm [34], and the overall mental workload of the multiple tasks [35, 36]. These evidences tend to suggest that these visual alterations could also affect driving.

The aim of this study is to investigate how a degradation in visual quality induced by a myopic defocus impairs driving behavior, in particular when the task workload is high. The experiment consisted of a driving simulator task in which young adults had to complete two scenarios (i.e. rural and highway) with distinct cognitive demands. In the rural scenario, participants performed a single driving task—inducing lower task workload—whereas in the highway scenario, they drove on a road while simultaneously engaging in a visual search task —inducing higher task workload—in their periphery, adjacent to the dashboard. Visual quality degradation was induced by means of contact lenses with specific additive power in order to achieve two different levels of visual acuity reduction. In both scenarios, the driving behavior of participants was analyzed as a function of the different visual conditions. We hypothesized that participants with visual quality degradation would show greater impairment of their driving behavior when both visual and cognitive demands are the highest (i.e., during the navigation task in the highway scenario) and that this effect would be more important with a higher visual quality degradation. Both speed [37–43] and SDLP [44–47] are widely studied variables in order to understand people's reactions to driving. We propose the hypothesis that the speed will be modulated by an increase in its variability as well as a decrease in its average and that SDLP will undergo an increase.

## Materials and methods

### Participants

A total of 21 French speakers' volunteers (5 women, 16 men), between 21 and 34 years old (mean ± SD = 24.8 ± 3.7) were recruited at the Université de Montréal (Québec, Canada). All participants gave their informed written consent prior to the experiment which conformed to the Declaration of Helsinki (2013). Experimental procedures were approved by the health research ethics committee at Université de Montréal (Comité d'éthique de la recherche clinique (CERC); certificate N˚18-090-CERES-D).

All participants had normal or corrected-to-normal vision and were in good general health. In addition, they all had a valid driver's license for at least five years (maximum 16 years). They met the legal criteria for driving in Québec, Canada: having a field of view of at least 100 continuous degrees along the horizontal meridian, 10 continuous degrees above fixation, and 20 continuous degrees below fixation with both eyes open and examined together (Légis Québec; Statutes of Quebec, 2015) and far visual acuity greater than or equal to 6/15 according to the Early Treatment Diabetic Retinopathy Study scale (ETDRS) located at 5 meters. Participants were recruited based on their degree of ametropia: myopia and bilateral hypermetropia severity had to be lower than -3 or +3 diopters (D), respectively. Other refractive errors had to be inferior or equal to 0.75D for astigmatism and 1.00D for anisometropia. To take part in the

experiment, participants with a refractive error were asked to wear their usual contact lenses to be corrected to normal. For logistical reasons in connection with eye tracking analyzes not covered in this thesis, participants with a refractive error wearing only glasses were not selected for the study.

## Materials

**Apparatus.** The Virage VS500M car driving simulator (Virage Simulation Inc.®, Montréal, Canada) was used for the driving tasks. Participants were seated in a faithful reproduction of a driver's cabin with a dashboard, a steering wheel, pedals, ventilation and a gearbox. The visualization system is composed of a generator 5-channel PC images and three 52-inch high-definition rear-projection screens (1280 × 720 pixels), thus providing the driver with a 180˚ field of vision. It also includes two side screens, behind the driver's seat, representing the blind spots. Finally, interior and exterior mirrors are integrated into the main front screens. A three-axis platform with electric cylinders was used to mimic the motion as well as the vibrations of a car by simulating the effect of accelerations, braking, but also vibrations generated by the engine and the tire contact on the road. The driving simulator also included a high-fidelity 5.1 stereophonic sound system which correlated with road conditions such as speed. Furthermore, additional realism was provided by the simulation of the Doppler effect which allowed for the recreation of noises generated by surrounding road traffic. In addition to the driving task, participants had to perform a visual search task (Fig 1) on a navigation device. An 8-inch Neewer NW801H monitor was used for the visual search task. The navigation device was positioned at the right of the participants adjacent to the dashboard at the level of the car's ventilation system. Participants were seated with their eyes at an approximate distance of 60 to

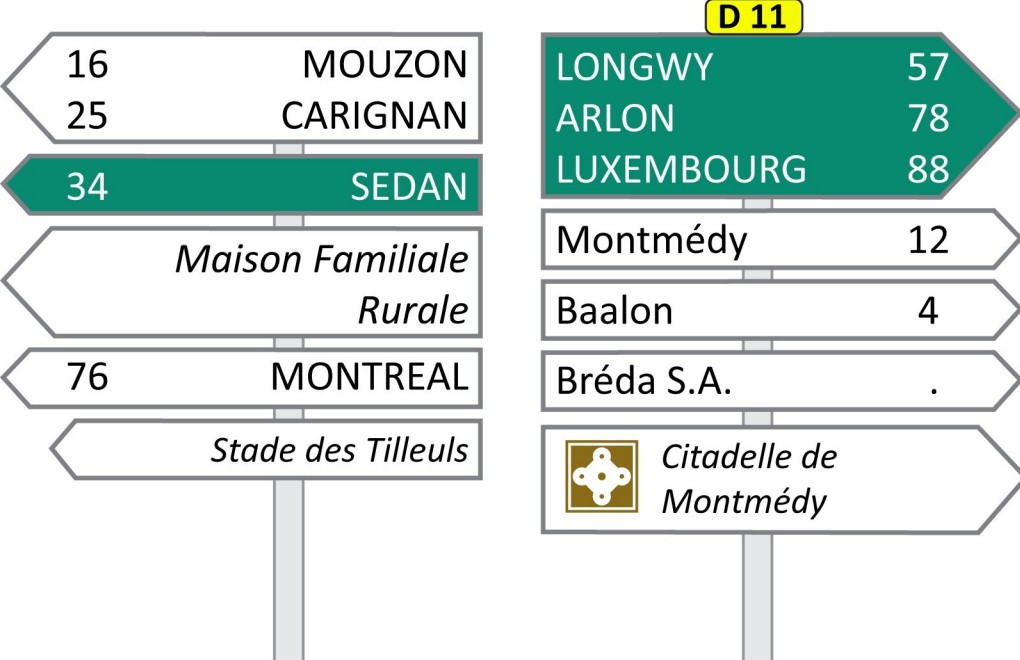

**Fig 1. Example of a visual stimulus displayed on the navigation device during the secondary task.** Multiple city names, with no semantic link, and in different typographies, both to avoid prediction from the participants, were displayed different types of road direction signs. The participant had to detect the city "Montreal" and say the associated number out loud (76 in this example).

70 cm from the device, according to participant's comfort, corresponding to an average distance between a navigation tool and the human eye. The monitor provided 160 degrees viewing angle with a standard 1024x768 pixel resolution and 500:1 contrast level. The images presented to the participants fitted the navigation device dimensions. The names of cities associated with highway exits as well as the numbers of these exits were written in size 15 Calibri font. The target word of the visual search ("Montreal"), was therefore 2.50 cm wide by 0.50 cm high.

**Experimental design.** The rural scenario was designed to simulate a straight road between and through small towns during a sunny day. It includes various hazardous events such as pedestrians crossing the street unexpectedly or cars infringing traffic regulation. Participants had to drive as they would in real life, obeying speed limits, road signs and considering other road users. At the beginning of the experiment, participants were told that they would encounter three different speed limits (90, 70 and 50 km/h) depicted on road traffic signs during the driving. The speed constraint was imposed so as to avoid potential compensation strategies such as a reduction in naturally adopted speed by some participants [26]. In order to help them monitor their speed, the driving simulator was programmed to produce a high pitch sound when participants were driving too fast and a low pitch sound when they were driving too slowly. In both scenarios, the participants were free to choose their preferred position with respect to the traffic lane.

The highway scenario consisted of a simple primary driving task, also on a straight road, paired with a secondary task displayed on a navigation device. The presence of these two concurrent tasks imposed greater visual and cognitive demands on participants, thus resulting in a higher task workload. The driving scenario has been designed to reproduce the insertion and exit of a highway. At the beginning of the experiment, participants were instructed to maintain a speed of 90 km/h as accurately as possible and the speed limit also appeared on road traffic signs during the driving. The secondary task consisted of a visual search task presented on a GPS navigation tool, located in the periphery on the car center console. This task was used in order to get participants to reallocate their attention between the road and the navigation device throughout the scenario, therefore challenging visual and attentional resources. The visual search task comprised 7 different visual stimuli depicting road direction signs with several pieces of information and city names (Fig 1). The presentation order of the GPS events, each lasting 6 seconds, was randomized using Matlab software. Participants had to find the number of the exit associated with the city "Montreal" and utter it out loud to the experimenter. Participants were permitted to answer both while the stimuli were present or after they disappeared and also to correct their response in case of self-recognized error.

## Protocol

For each participant, the experiment lasted around 2h30m. As illustrated in Fig 2, participants first did visual pre-tests including ETDRS for visual acuity, Randot for stereoacuity and Humphrey for visual field. They then performed two blocks, each one consisted of an initial familiarization phase with the driving simulator (as recommended by numerous sources [48, 49]) and two driving scenarios (highway and rural), followed by a Simulator Sickness Questionnaire (SSQ). Following a within-subject design, all participants did one block under optimal vision and another one under degraded vision. Therefore, each participant performed both driving scenarios (rural and highway) in each of the visual conditions (optimal and degraded vision), in a randomized and counterbalanced order (Fig 2). The two versions of the driving scenarios, presented in optimal and degraded vision, differed regarding when stimuli (hazardous events in rural and GPS events in highway) were presented throughout the task. This

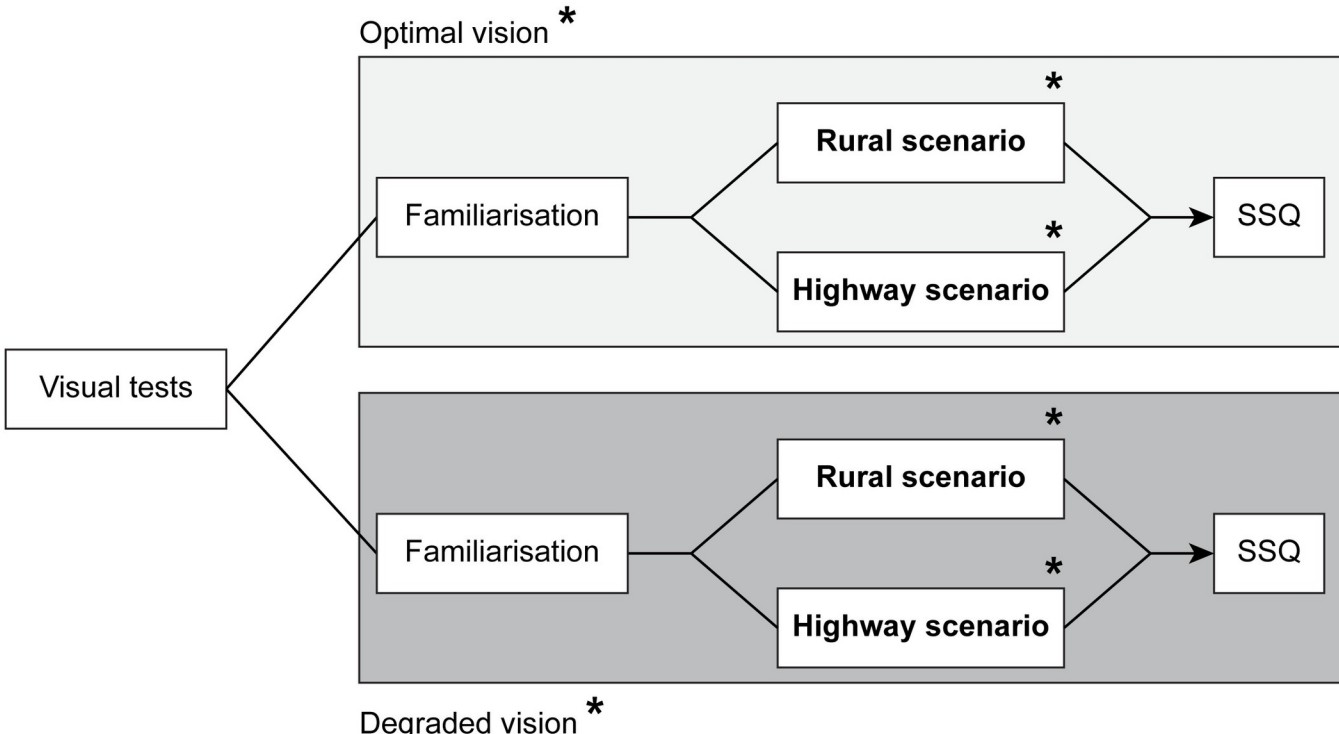

**Fig 2. Protocol.** Example of an experimental sequence performed by participants. The order of the visual conditions (optimal and degraded vision) and the driving scenarios (rural and highway) was counterbalanced across participants, as depicted by the asterisks.

ensured that these events remained unpredictable to participants. The first familiarization—regardless of the order of the acuity condition and the scenarios performed—was 10 minutes long. The second was shorter, representing an additional 2 minutes of driving. The choice of the familiarization duration was influenced by the work of [50], who proposed that 5 to 10 minutes of driving in a simulator are effective in promoting the comfort of the participants.

## Visual degradation

The 21 participants were divided into two different groups according to a targeted visual quality reduction achieved by the use of contact lenses. This manipulation was executed in order to examine the effect of different levels of deterioration in the quality of vision on driving behavior. The first group of 11 participants (24.2 ± 3.5 years old), entitled "lower degradation" group, had a visual quality degradation to 6/15 corresponding to the legal visual standard for driving in Québec, Canada. The second group of 10 participants (25.5 ± 3.9 years old), entitled "higher degradation" group, had a visual quality degradation equal to 6/75. This acuity threshold was chosen in order for participants to get blurred vision at the approximate distance of the simulator screen (120 cm). Distance between the participant and the simulator screen could vary by a few centimeters since the latter adjusted the distance from the driver's seat according to their comfort and according to their size. The reduction in acuity was achieved by an optometrist through the use of the ETDRS optometric test. For the degraded visual conditions, the power of the contact lenses was defined by calculating the difference between the target visual acuity threshold and the visual acuity of the participant, assessed without correction. The additive power of the contact lenses was then inducing a myopic defocus resulting in a decrease in visual acuity. Daily disposable contact lenses were used and discarded after each

experimental session in order to prevent any risk of contamination. Contact lenses were removed at any time if any discomfort was experienced by the participant.

## Driving measures and data analysis

Driving behavior was assessed by seven driving variables, divided into three categories, as shown in Table 1. Those variables have previously been identified as significant and non-redundant driving behavior measures for the rural scenario [26]. For each data point, speed equal to 0 km/h, inferior to 10 km/h or recorded 100m before or after an event were discarded from the averaging. Additionally, for each data point, lateral position recorded 10 seconds before and after a lane changing were discarded from the averaging. Mean speed was computed using a simple average formula (i.e. MS = mean[X]) in which X as the valid data after exclusions. Speed variability was computed using $SV = \sqrt{(\text{mean}(X-\text{mean}(X))^2)}$. Standard deviation of the lateral position (SDLP) was computed using 2 steps, first to obtain the mean lateral position (LP) (i.e MLP = mean[LP]) and second to obtain the SDLP (i.e. $SDLP = \sqrt{(\text{mean}(LP-\text{mean}(LP))^2)}$).

Three driving variables were analysed in the highway scenario: mean speed, speed variability and SDLP. Definitions of the metrics remain the same for both scenarios, but exclusion and computation of data were slightly different. Data from the three metrics were excluded if greater or less than 3 standard deviation from the mean. The standard deviations of each metric were first calculated for each GPS event and then averaged across all seven events. Unlike the rural scenario, participants had to perform a secondary visual search in addition to the primary driving task (i.e., double task). The secondary task was presented seven times throughout the scenario (Fig 3, correct and incorrect responses are represented by green and red shaded areas, respectively). The duration of each GPS event was 6 seconds and a random delay of several seconds was introduced in between two consecutive presentations. A reference period of 60 seconds, was used as a baseline to assess driving behavior without interference of the secondary task—thus, single task. The reference was initially divided into two 30-s time windows: one before the first GPS event and another one after the last GPS event (Fig 3, blue shaded areas). Start and end times as well as the length for each window were adjusted in order to

**Table 1. Short version of Michaels et al., (2017) studied measures definitions and units in which they were recorded (5).**

| Category | | Measure | Unit | Description |
|---|---|---|---|---|
| **Classical driving variables** | 1 | Crash | n | Number of crashes per experimental condition. |
| | 2 | Near crash | n | Number of near crashes per experimental condition. When within an event:<br>• Subject brakes harder than a given threshold (0.7) while driving at a speed greater than 18 km/h<br>• The steering wheel is turned more than 60 degrees while driving faster than a speed threshold (18 km/h)<br>• The participant drives within 3 m of an object while travelling at a speed greater than 36 km/h |
| | 3 | Mean Speed* | km/h | Average speed of all driving in each experimental condition. |
| | 4 | Speed variability* | km/h | Standard deviation of the speed in each experimental condition. |
| | 5 | SDLP* | m | Standard deviation of lateral position in each experimental condition. |
| **Abrupt or uncontrolled actions** | 6 | Max brake | n | Hardest amount of braking applied during events of interest in each experimental condition. Where 0 = no braking applied, 1 = pedal is fully depressed and where all values in between are possible scores. |
| **Anticipation while facing hazardous events** | 7 | Distance at max brake | m | Distance from object at which "Max brake" is recorded in each experimental condition. |

In this table, n corresponds to an undefined unity, m to meters, km/h to kilometers per hour

* to variables in the highway scenario.

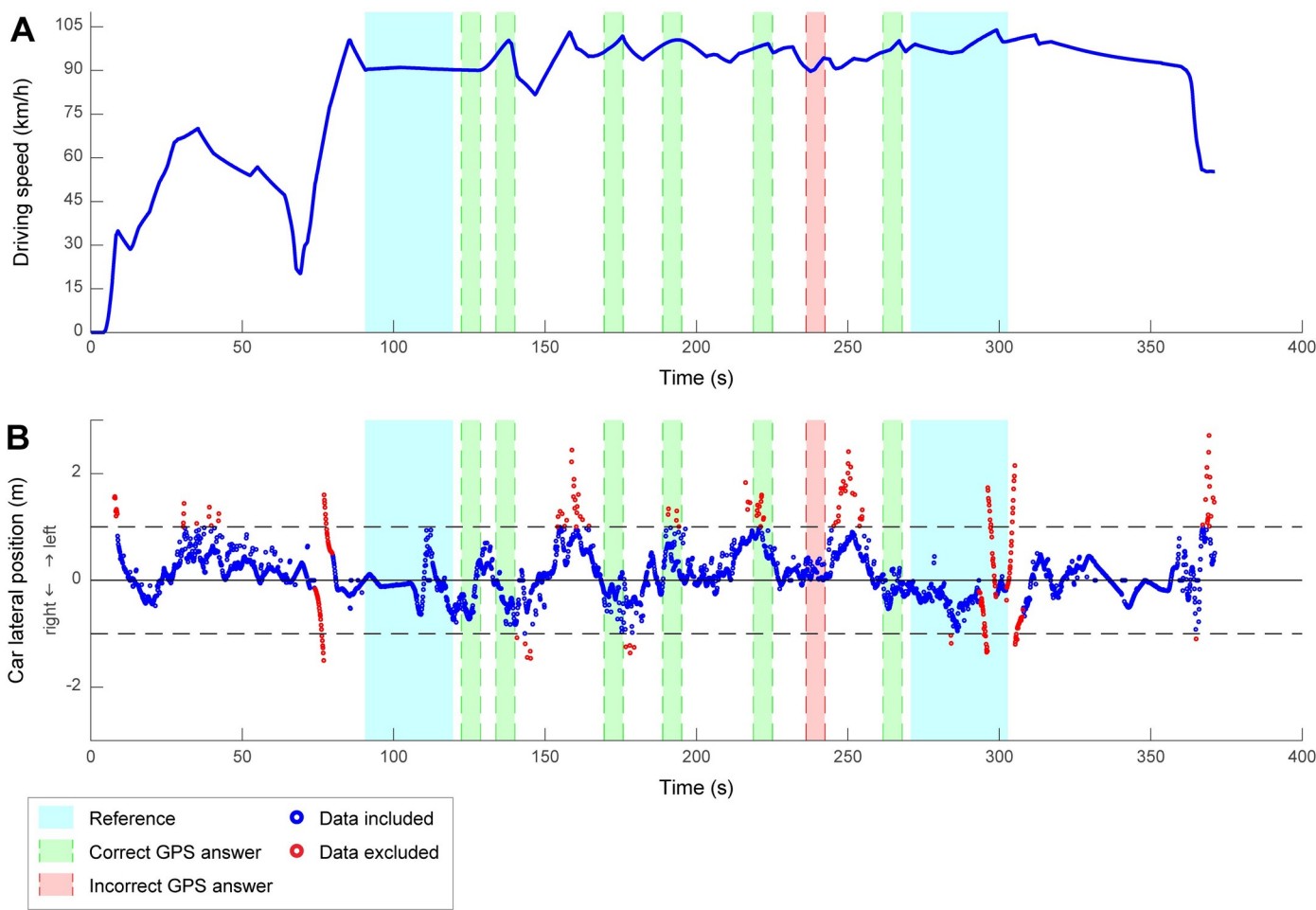

**Fig 3. Driving speed and car lateral position of a typical participant during the highway scenario.** (A) Driving speed (in km/h) as a function of time. At the beginning of the scenario (around 70 s), the car enters the highway, as shown by the increase in speed. Then, since the driver was instructed to keep a constant velocity of 90 km/h, the car speed stabilizes during the rest of the scenario and until the exit of the highway. (B) Car lateral position (in m) as a function of time. The center of the road corresponds to the solid horizontal black line, both sides of the road are represented by the dashed horizontal black lines and the blue and red dots represent the raw data.

discard time frames with highly variable speed (e.g., abnormal acceleration or abrupt braking) which ensured to have a 60-second reference period for every participant, with relatively stable driving speed. To compute the speed variability and the SDLP during the reference, standard deviations were computed for both time windows separately and were then averaged. For both driving scenarios, data were excluded if the participant had a velocity inferior to 70 km/h or 100 meters before and 100 meters after a detected crash or near crash. The data processing and the computation of the driving variables were performed using a custom-written Matlab® toolbox (The MathWorks, Natick, MA, USA).

## Statistical analysis

Firstly, the discomfort experienced by participants in the driving simulator was measured. The three SSQ subscores (nausea, oculomotor symptoms and disorientation) and the total SSQ scores were analyzed by means of two-way mixed ANOVAs with visual condition (optimal or degraded vision) as a within-subjects factor and degradation group (lower and higher

degradation) as a between-subjects factor. The data of two participants were excluded from the SSQ analysis, due to an inability to collect data, allowing a total of 19 participants to examine the discomfort in the simulator. The rest of the analysis included a total of 21 participants.

Secondly, the driving behavior in both rural and highway scenarios was assessed to evaluate the effect of increased visual and cognitive demands. In the rural scenario, seven different variables (see paragraph 2.5) were used to describe participants' driving behavior. Those variables were analyzed using two-way mixed ANOVAs with visual condition (optimal or degraded vision) as a within-subjects factor and degradation group (lower and higher degradation) as a between-subjects factor.

Finally, in the highway scenario, the behavior on the secondary task (i.e. visual search task displayed on a GPS navigation tool while driving) was measured by the success rate, which corresponds to the proportion of correct responses. Success rate on the secondary visual search task was analyzed using two-way mixed ANOVAs with visual condition (optimal or degraded vision) as a within-subjects factor and degradation group (lower and higher degradation) as a between-subjects factor. Three driving variables were measured in the highway scenario (see paragraph 2.5) and were subject to two-way analyses of covariance (ANCOVAs). The ANCOVAs allowed testing for the effects of visual condition (optimal or degraded vision) and degradation group (lower and higher) on the driving variables, while considering the cognitive load (single versus double task) as a covariate. This, in order to reduce the variability of response at the double task, bring out the visual effect. In the case where variables did not follow a normal distribution, Box-Cox transformations were applied in order to adjust for skewness in the data (Table 2). Statistical thresholds were set at P < 0.05.

## Results

### Discomfort in the driving simulator according to acuity conditions

Two-way mixed ANOVAs were performed in order to compare the visual quality conditions as well as the degradation groups. As illustrated in Fig 4, no significant difference was observed for the nausea subscores between visual conditions (optimal vision: 26.50 ± 43.51, degraded vision: 23.85 ± 27.78) or degradation groups (lower degradation: 19.08 ± 37.31, higher degradation: 28.62 ± 35.15) (all p > 0.05) and no interaction was observed between the visual conditions and degraded groups (all p > 0.05). The analysis showed significant increases in scores under reduced vision quality for oculomotor (optimal vision: 16.00 ± 16.93; degraded vision: 42.95 ± 22.58, $F(1,34) = 21.577$; $p < 0.001$; $\eta^2 = 0.376$), disorientation (optimal vision: 29.39 ± 48.22; degraded vision: 64.96 ± 49.17; $F(1,34) = 16.48$, $p = 0.0003$, $\eta^2 = 0.304$) as well as

**Table 2. Driving behavior, secondary task and simulator sickness variables normality.**

| Driving behavior | Driving behavior | Simulator sickness |
|---|---|---|
| In the rural scenario | In the highway scenario | |
| Crashes* | Speed | Nausea* |
| Near crashes* | Speed variability* | Oculomotor* |
| Max brake | SDLP* | Disorientation* |
| Distance at max brake* | | Total* |
| Speed | Success rate* | |
| Speed variability | | |
| SDLP* | | |

* representing that the variable underwent a Box Cox transformation.

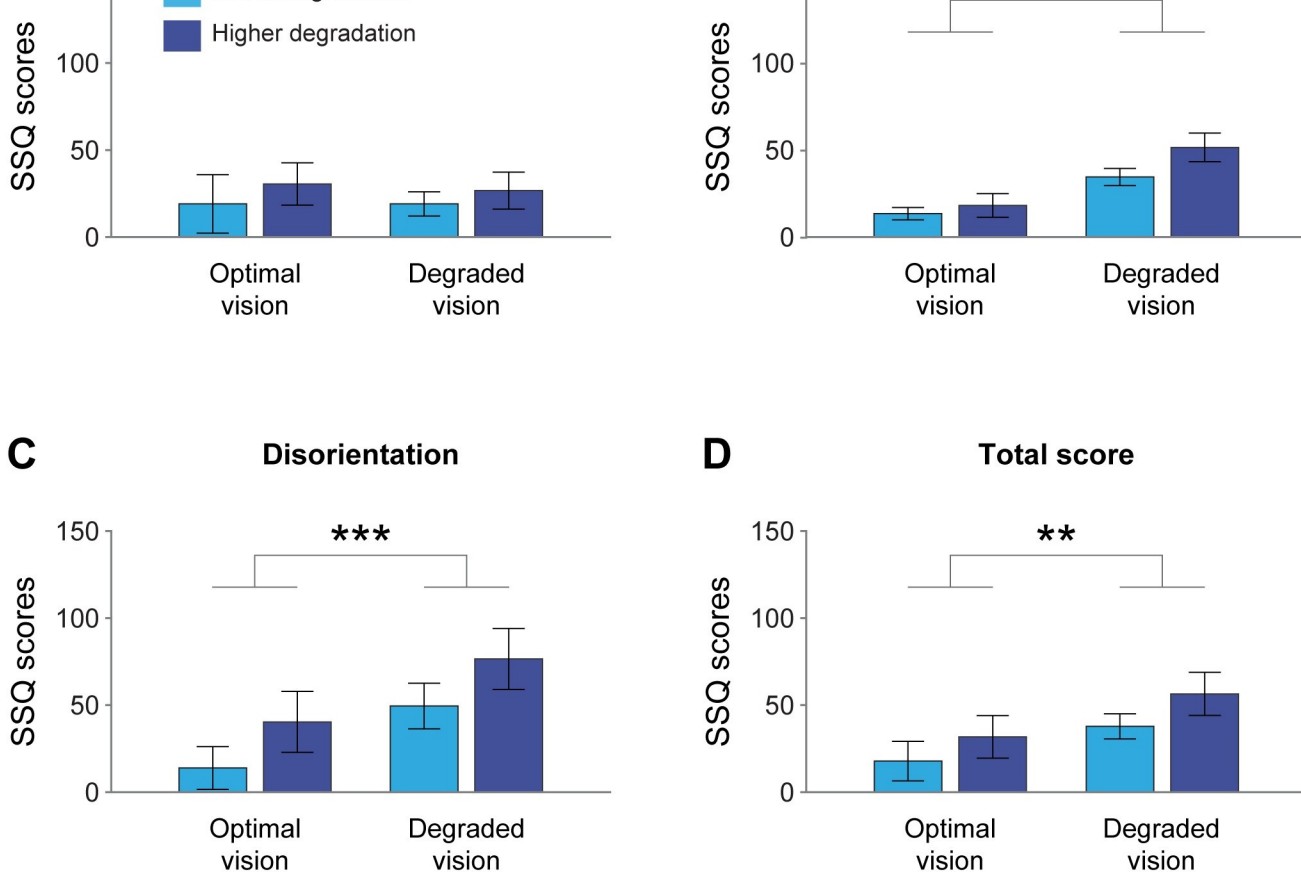

**Fig 4. Mean SSQ scores (± standard error) for the visual quality conditions and degradation groups (n = 19 participants).** (A) Nausea subscores (B) Oculomotor symptoms subscores (C) Disorientation subscores (D) Total scores. $^{**}$p < 0.01, $^{***}$p < 0.001.

total SSQ scores (optimal vision: 26.18 ± 36.32; degraded vision: 48.00 ± 32.70; F(1,34) = 10.674; p = 0.002 $\eta^2$ = 0.231). There was no significant main effect of the degradation group and no significant interactions (all p > 0.05) for these three SSQ variables (oculomotor subscore, disorientation subscore and total score). These results highlight perceived discomfort by the drivers while driving with degraded visual quality.

### Influence of reduced acuity according to visual and cognitive workload

**Rural scenario: Low visual and cognitive workload.** Table 3 summarizes the mean of each driving variable in the rural scenario. In order to test the effect of visual quality condition, two-way mixed ANOVAs were performed for each of the seven driving metrics with the degradation groups as categorical factors. The ANOVAs did not demonstrate any significant difference between visual conditions (all p > 0.05) nor between the degradation groups (all p > 0.05) for all the metrics and no significant interaction was found between visual condition and degradation group (all p > 0.05). These findings suggest that the driving behavior of participants was not significantly impaired by the deterioration of vision quality even at different

**Table 3. Mean (± standard deviation) driving behavior calculated for crash, near crash, mean speed, speed variability, SDLP, maximum brake and distance at maximum brake in optimal and degraded vision conditions for both degradation groups.**

|  | Optimal vision | | Degraded vision | |
|---|---|---|---|---|
|  | Lower degradation | Higher degradation | Lower degradation | Higher degradation |
| Crash (n) | 1.36 (± 0.92) | 1.30 (± 0.82) | 1.45 (± 0.82) | 1.50 (± 0.97) |
| Near crash (n) | 1.45 (± 0.69) | 0.80 (± 0.42) | 1.18 (± 0.60) | 1.20 (± 0.79) |
| Mean speed (km/h) | 68.69 (± 6.18) | 67.68 (± 4.63) | 69.65 (± 4.24) | 67.13 (± 4.08) |
| Speed Variability (km/h) | 16.77 (± 1.94) | 17.63 (±1.76) | 16.53 (±0.97) | 17.50 (±1.28) |
| SDLP (m) | 0.28 (± 0.06) | 0.24 (± 0.05) | 0.26 (± 0.06) | 0.23 (± 0.05) |
| Max brake (n) | 0.60 (± 0.25) | 0.62 (± 0.19) | 0.58 (± 0.13) | 0.71 (± 0.15) |
| Distance at max break (m) | 127.75 (± 87.02) | 127.14 (± 67.18) | 142.75 (± 60.54) | 102.24 (± 78.50) |

intensities in a context associated with low visual and cognitive demands. Our results do not indicate a decreased mean driving speed or increased number of crashes (Table 4) even at the higher visual quality degradation level. In order to better understand if the decreased visual acuity may impact driving behavior, we examined it under more demanding circumstances: the highway scenario.

**Highway scenario: High visual and cognitive workload.** The success rate on the secondary task was evaluated at 92.21% ± 9.82 for the lower degradation group and 85.71% ± 26.94 for the higher degradation group in the optimal vision condition and at 71.43% ± 24.74 for the lower degradation group and 88.57% ± 18.81 for the higher degradation group in the degraded vision condition. The variation between the visual quality conditions as well as the degradation groups was examined by a two-way mixed ANOVA (Fig 5). There was no significant main effect of the visual condition ($F_{(1,38)}$ = 2.435, $p$ = 0.127), no significant main effect of the degradation group ($F_{(1,38)}$ = 1.453, $p$ = 0.235) and no interaction between visual condition and degradation group ($F_{(1,38)}$ = 2.654, $p$ = 0.112). The success rate for the secondary task remained generally very high unaffected by visual and cognitive demands.

In order to examine the effect of visual demand when controlling for the variance of cognitive load (i.e., single task vs double task) in the highway scenario, two-way ANCOVAs, comprising visual quality condition and degradation group as factors and cognitive load as a covariate, were performed on the measured driving variables. This analysis allowed us to explore if there were any difference between or among visual conditions and degradation groups due to visual load. The following ANCOVAs analysis therefore report changes or stability of driving behavior in the context of double task in opposition to the single task behavior discussed in the previous section. Table 5 summarizes the means and standard deviations of the mean speed, speed variability and SDLP in each experimental condition.

For mean speed, a two-way ANCOVA (Fig 6) did not show a main effect of the covariate cognitive load ($F_{(1,79)}$ = 0.42, $p$ = 0.519) or a main effect of the visual quality conditions adjusted for the cognitive load covariate ($F_{(1,79)}$ = 0.85, $p$ = 0.360). However, it revealed a significant main effect of the degradation groups adjusted for the covariate ($F_{(1,79)}$ = 4.26, $p$ = 0.042, $\eta^2$ = 0.05; grand means lower degradation group = 90.10 km/h ± 4.43, grand means

**Table 4. Frequencies of crashes and near crashes in optimal and degraded vision conditions for both degradation groups.**

|  | Optimal vision condition | | Degraded vision condition | |
|---|---|---|---|---|
|  | Lower degradation | Higher degradation | Lower degradation | Higher degradation |
| Crashes | 15 | 13 | 16 | 15 |
| Near crashes | 16 | 8 | 13 | 12 |

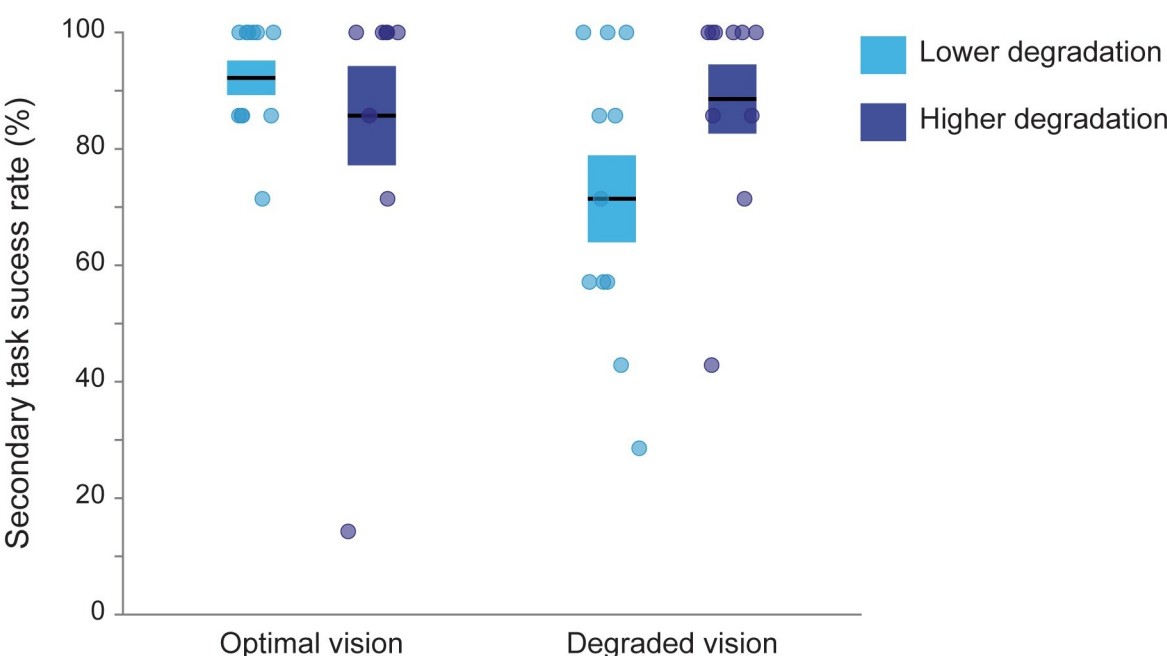

**Fig 5. Success rate on the navigation device visual search task according to visual quality condition and degradation group.** Boxes represent the mean success rate ± standard errorsto the mean and circles correspond to individual data.

higher degradation = 88.11 km/h ± 4.32). No significant interaction was observed between the visual conditions and the degradation groups while adjusting for the covariate (F(1,79) = 0.38, p = 0.538). Thus, mean speed was only affected by a high level of visual degradation when controlling for the variance of the cognitive load.

For the speed variability, as represented in Fig 7, a two-way ANCOVA revealed a main effect of the covariate cognitive load (F(1,79) = 74.80, p < 0.001; $\eta^2$ = 0.46; grand means single task = 2.75 km/h ± 0.92; grand means double task = 5.49 ± 2.35). The ANCOVA also revealed a main effect of the visual quality condition (F(1,79) = 5.21, p = 0.025, $\eta^2$ = 0.03, grand means optimal vision = 3.76 km/h ± 2.07, grand means degraded vision = 4.48 ± 2.38) and the degradation groups (F(1,79) = 4.11, p = 0.046, $\eta^2$ = 0.03; grand means lower degradation group = 3.85 km/h ± 2.19, grand means higher degradation group = 4.41 km/h ± 2.30) when adjusted for the covariate cognitive load. No interaction was observed between the visual conditions and the degradation groups when adjusting for the covariate cognitive load (F(1,79) = 0.002,

**Table 5. Driving behavior variable means (± standard deviations) for each visual condition and degradation group according to the level of cognitive load.**

|  | Single task | | | | Double task | | | |
|---|---|---|---|---|---|---|---|---|
|  | Optimal vision | | Degraded vision | | Optimal vision | | Degraded vision | |
|  | Lower degradation | Higher degradation | Lower degradation | Higher degradation | Lower degradation | Higher degradation | Lower degradation | Higher degradation |
| Mean speed (km/h) | 88.76 ± 4.11 | 88.80 ± 3.62 | 91.68 ± 4.04 | 88.45 ± 2.80 | 89.9 ± 4.79 | 87.12 ± 4.20 | 90.14 ± 4.70 | 88.06 ± 6.38 |
| Speed variability (km/h) | 2.37 ± 0.66 | 2.82 ± 1.07 | 2.71 ± 1.07 | 3.14 ± 0.78 | 4.53 ± 1.97 | 5.38 ± 2.57 | 5.89 ± 2.61 | 6.30 ± 2.20 |
| SDLP (m) | 0.27 ± 0.06 | 0.26 ± 0.05 | 0.26 ± 0.06 | 0.23 ± 0.07 | 0.42 ± 0.12 | 0.38 ± 0.12 | 0.45 ± 0.07 | 0.42 ± 0.09 |

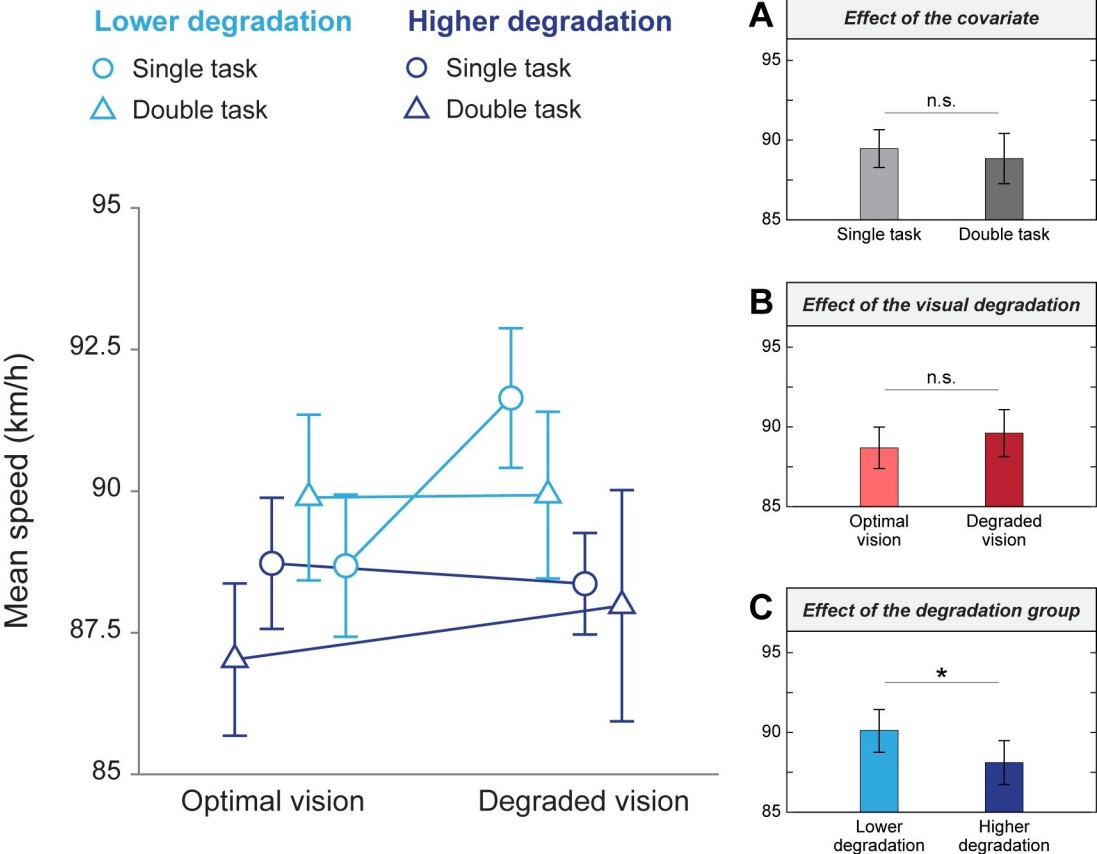

**Fig 6. Mean speed as a function of visual quality conditions and degradation groups.** Mean values from simple and double tasks are represented by blue circles and triangles, respectively. Different shades of blue represent the lower (lighter) and higher (darker) degradation groups. Error bars correspond to the standard error of the mean. (A) Means in single and double tasks. (B) Means in optimal and degraded vision, unadjusted for the covariate. (C) Means in the lower and higher degradation groups, unadjusted for the covariate. $^{*}$p < 0.05, n.s.: non-significant.

p = 0.962). These findings show that speed variability is affected by the cognitive load, and, in addition to this, by statistically controlling the impact of cognitive load, the effect of visual load emerged in the dual task context.

As depicted in Fig 8, a two-way ANCOVA revealed a main effect of the covariate cognitive load on the SDLP ($F_{(1,79)} = 79.94$, $p < 0.001$; $\eta^2 = 0.50$; grand means single task = 0.26 m ± 0.06; grand means double task = 0.42 m ± 0.10) only. No main effect of the visual quality condition ($F_{(1,79)} = 0.10$, $p = 0.755$) or the degradation groups ($F_{(1,79)} = 2.01$, $p = 0.160$) were observed when adjusting for the covariate cognitive load. Similarly, there was no interaction between the visual conditions and the degradation groups when adjusting for the covariate cognitive load ($F_{(1,79)} = 0.25$, $p = 0.621$). These results indicate that SDLP is increased when the cognitive load is more important, but is not affected by the visual degradation induced.

## Discussion

The aim of this study was to determine whether a visual quality degradation inducing a myopic defocus and other visual issues had a greater impact on driving behavior and drivers' comfort

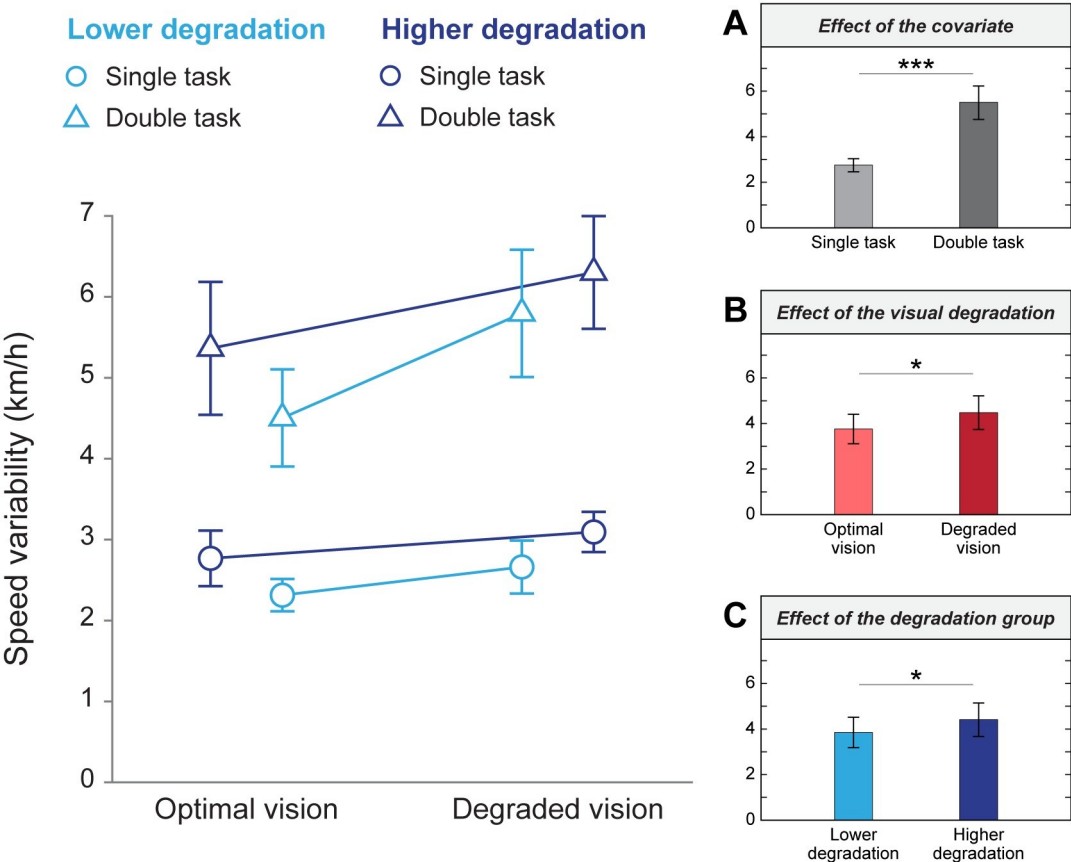

**Fig 7. Speed variability as a function of visual quality conditions and degradation groups.** Mean values from simple and double tasks are represented by blue circles and triangles, respectively. Different shades of blue represent the lower (lighter) and higher (darker) degradation groups. Error bars correspond to the standard error of the mean. (A) Means in single and double tasks. (B) Means in optimal and degraded vision, unadjusted for the covariate. (C) Means in the lower and higher degradation groups, unadjusted for the covariate. $^*p < 0.05$, $^{***}p < 0.001$.

when participants were engaged in multiple tasks involving high interaction between visual and cognitive mechanisms. Twenty-one participants performed a simple driving simulator task (rural scenario) and a driving task paired with a visual search task (highway scenario). Their driving behavior was examined in optimal vision and with degraded visual quality. Participants were divided into two groups with distinct thresholds of reduced visual acuity: the lower degradation, which corresponded to the minimum visual acuity required to drive in Québec, Canada and the higher degradation, which resulted in blurred vision at the distance of the front screen of the driving simulator. In the rural scenario—associated with lower visual and cognitive demands—, our results revealed that the reduction in visual acuity did not impair young participants' driving behavior nor their ability to drive safely. On the other hand, the highway scenario—associated with greater workload and visual demands—appeared to have elicited an effect of visual degradation on driving behavior. To summarize, our study emphasizes three main results: i) the perceptual and motor incoherence induced by the experimental disturbance of vision as an enhancer of evaluated motion sickness symptoms suggesting greater discomfort, ii) an impact of reduced visual quality in the presence of a double task and iii) an effect of visual acuity higher degraded.

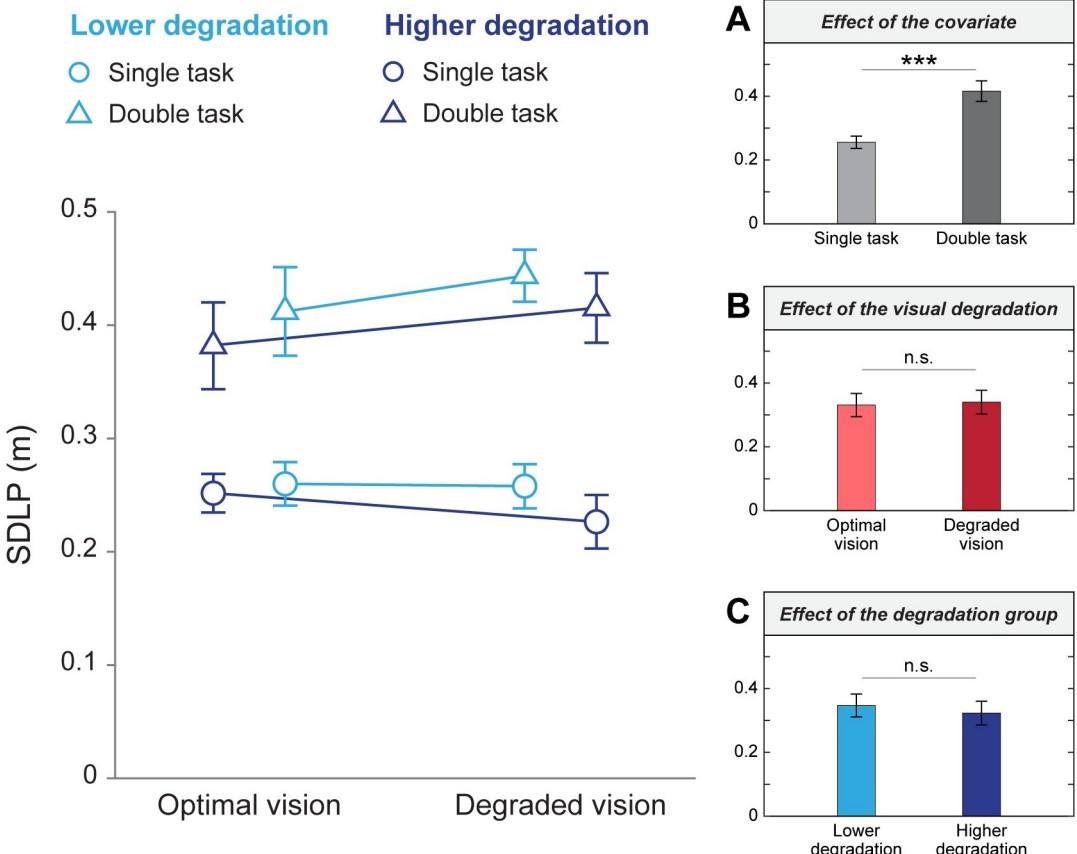

**Fig 8. SDLP as a function of visual quality conditions and degradation groups.** Mean values from simple and double tasks are represented by blue circles and triangles, respectively. Different shades of blue represent the lower (lighter) and higher (darker) degradation groups. Error bars correspond to the standard error of the mean. (A) Means in single and double tasks. (B) Means in optimal and degraded vision, unadjusted for the covariate. (C) Means in the lower and higher degradation groups, unadjusted for the covariate. ***p < 0.001, n.s.: non-significant.

### Perceptual incoherence provided by the experimental visual inputs disturbance as an enhancer to evaluate motion sickness symptoms suggesting greater discomfort

The degradation of visual quality by means of contact lenses with positive addition had a negative impact on participants' comfort while being in the driving simulator, as shown by greater disorientation, oculomotor symptoms, and total SSQ scores. Considering that some variability of this effect could be related to the physical disturbance provoked by the contact lenses, we hypothesize that most of the effect is driven by the occurrence of the changes of accommodation convergence and non-corresponding retained images. The weight of the impact of both optical, induced by the myopic defocus, and physical disturbance on simulator sickness is however still to be elucidated. The idea of changes of accommodation convergence and non-corresponding retained images as driving the impact of myopic defocus rather than physical discomfort is reinforced by the deterioration in driving behavior, only observed in one of the two driving scenarios. Indeed, the analysis of the driving metrics revealed a deterioration in driving behavior as a function of the workload and the level of visual degradation measured in the highway scenario. We then may assume that the increase in motion sickness is driven by this scenario, which involves higher visual, i.e. accommodation-convergence, changes in

distances, increased demand in high frequency resolution for the visual search task on the navigation device, motor, i.e. eye, head and neck motor coordination and transitions between the road and the navigation device, and cognitive demands i.e. visual search task compared to the rural scenario. It has been previously suggested that increased cognitive load was not associated with simulator sickness [51]. However, the cognitive demands in the highway scenario are explained by the addition of a concurrent visual search task, involving a specific type of workload ultimately resulting in increased ocular motricity and visual demands. In fact, participants had to switch their visual and attentional focus between the road in far distance and the navigation device in intermediate distance. It has been shown that depth perception relies on multiple visual cues but also oculomotor information emerging from the accommodation responses to focus from one distance to another [52]. In our study, the myopic defocus induced in our participants gave rise to a sensorimotor conflict between the retinal image, the accommodation and vergence response and the extra-retinal perception based on oculomotor commands and the proprioception of accommodation. It is also well known that the alteration of the accommodation and convergence mechanisms has negative consequences on task performance as well as the visual fatigue experienced [53]. Our results suggest that sensorimotor conflict, as a consequence of visual perturbation, results in oculomotor discomfort and disorientation in the driving simulator.

Moreover, the magnification induced by contact lenses creates a perceived motion of the visual scene that is different from the motion encoded by the semicircular canals in the vestibular system. There is extensive literature showing that motion sickness experienced in driving simulators is due to the discrepancy between the visual perception of movement and the absence of real movement [54, 55]. According to the sensory rearrangement theory, motion sickness is caused by conflicting motion signals processed through the vestibular, visual and somatosensory systems as well as the mismatch between the expected and actual stimuli [56]. It has been shown that, just like the vestibular system, the cerebellar mechanisms are involved in both sensory conflict by the motor control of the head and body motion and the associated corollary discharge signals and reafferent signals [57]. This shows that almost any variation in expected responses between sensory or proprioceptive mechanism can induce motion sickness, (cf. review from [54]), here visually induced by the contact lenses and the multiple visually and cognitively demanding tasks of driving. Finally, our findings are consistent with previous research investigating the effect of visual blur and motion sickness in optokinetic drum [58] or visual fatigue using virtual head-mounted displays [59, 60]. Discomfort was also perceived in visually induced motion sickness while adapting to spectacles [61] and visual aids [62].

## Impact of degraded visual quality in the presence of a double task

The rural scenario did not reveal any effect of the visual degradation nor the degradation group. This lack of significance may indicate that the task workload, the visual image processing or the interaction between visual and cognitive mechanisms was not challenging enough to impact the driving behavior of our participants. In addition, the absence of an increase in crashes and near crashes between the visual acuity conditions and the degradation groups emphasizes the well-known difficulty to assess accidentology causal relationships with visual acuity. Moreover, the main task of the rural scenario, the avoidance of unexpected events relies more on motion perception, rather than high spatial-frequency perception. In contrast, the double-task paradigm in the highway scenario, without consideration of the visual condition, appears to be more effective at eliciting modifications of the driving behavior that could potentially compromise road safety. In the highway scenario, our results revealed some modulations

of vehicle position in the mediolateral axis, captured by the SDLP, as well as in the anteroposterior axis, as expressed by the speed variability. Both SDLP and speed variability were increased during the double task in comparison to the single task. This finding is consistent with previous studies that reported greater variability in vehicle lateral position in tasks with high cognitive load using different metrics, namely lane position [47], lane keeping [44, 63, 64] and steering [65, 66]. As reported in previous literature, speed reduction is a common strategy adopted by drivers to compensate for increased mental workload when performing concurrent tasks such as reading texts messages while driving [42] or to compensate for decreased visual-cognitive abilities related to aging [26, 67]. However, this reduction in mean driving speed has been shown in studies including a double task on the real road, which is associated with actual crash risk [68]. Driving in a simulator is not associated with real-life consequences related to crash damage, which probably uninhibits young drivers in their driving behavior compared to their usual driving in real road conditions. In addition, speed reduction, as a driving strategy, was also observed during a single driving simulator task with older adults dealing with altered cognitive capacities [26]. This suggests that the impact of the double task on speed behavior, expressed by an increase in speed variability, could reflect a behavioral difference between younger and older drivers. Indeed, our participants were young healthy university students with functional cognitive capacities, but we might expect greater changes in driving behavior in older participants doing the highway task. Interestingly, most studies about driving have used the reallocation of attention and visual focus in depth and eccentricity to create a dual-task paradigm implying high-level visual processing. This has notably been put forward in the ISO standards as a basis for examining the driver's visual behavior with respect to the in-vehicle devices and the induced workload [69–71]. This methodology, along with low-level visual sensory integration mechanisms, as demonstrated by our results, seems promising in the analysis of driving behavior. This will be discussed in the next paragraphs addressing the impact of visual quality reduction in dual task context.

An additional metric that we used in the present study to measure the impact of visual acuity reduction on driving in the highway scenario was the success rate at the visual search task. Since no significant decrease in driving speed was observed in the presence of a double task and degraded vision, we could have expected lower success rates in the degraded vision condition, especially in the higher degradation group. However, although all participants subjectively reported a discomfort associated with blurred vision at the distance of the navigation device, no significant decrease in the task success rate was observed when vision was degraded. Furthermore, success rates did not significantly differ between the lower and higher degradation groups. It could be that the success rate is not a metric sensitive enough to indicate reduced efficiency or low-level integration cost resulting from the interference between the visual and cognitive mechanisms in young drivers. Similarly, a recent study investigated the effect of a simulated reduction in acuity on a pedestrian detection task and revealed no reduction in the success rate but the authors found that response times were lengthened by approximately 600 ms [72]. Moreover, it has been shown that when participants were asked to prioritize driving, they exhibited longer dialing times as well as better stabilization of the car lateral position on the road [73]. Therefore, it seems that the combined analysis of response time and success rate would provide a more accurate description of task performance. It is important to note that, in our experimental design, each road sign used in the visual search task was presented for 6 seconds on the navigation device and no time limit was imposed for participants to answer. Hence, it is possible that participants had enough time to correct their answers, which would explain the overall good success rates observed irrespective of the conditions. Moreover, it is very likely that they did several visual back-and-forth between the navigation device and the road during the task, as fixations have been shown to last around 120 ms

[74, 75] but can be as short as 26 ms [76]. As a consequence, it might have helped participants to maintain a stable SDLP and give correct answers, despite the reduction in visual quality. Given the spatial frequency of the target city name, it is very unlikely that the reduction in visual acuity has disturbed the integration of the image at an intermediate distance, especially when considering the accommodative capacity of our young population. However, despite the fact that we did not observe any impact on the success rate, the driving measures suggest a loss of efficiency due to the visual degradation.

Indeed, our results indicate that the highway scenario and the concurrent visual search task led to modifications of the driving behavior when the quality of vision was degraded. As previously mentioned, the reduction in visual acuity resulted in visual blur, but also altered the accommodation and convergence relationship, which is known to be associated with refractive error [77–79]. It remains unknown whether visual demands in intermediate and far distances are key elements to induce such an effect; however, the solicitation of visual quality alteration and increased workload seems to provide a relevant methodology to assess loss of efficiency while driving. In fact, several authors studying various contexts such as a Stroop task [80] and a bingo task [81] suggested that visual alteration could alter the performance in paradigms involving a high cognitive demand. The information degradation hypothesis suggests that certain higher order mistakes can occur following errors in perceptual processing, which can be explained by degraded perceptual inputs [82]. In contrast to what has been reported in previous studies [17, 18], we did not show any significant effect of the visual perturbation on the SDLP in the context of high cognitive load. Nevertheless, this result is in line with other findings showing refractive blur did not significantly affect road position [16] and steering wheel position [14, 17, 18]. Our results showed that the driving speed variability was increased not only in the dual task when compared to the simple task but also in the degraded visual quality condition compared to optimal visual quality condition. We were thus able to demonstrate the impact of an alteration in low-level visual mechanisms on driving behavior in a context of dual task in which the impact of cognitive load was controlled, therefore permitting the emergence of the impact of visual load on driving behavior. We hypothesize that, because of the increase in visual demands associated with altered visual quality, participants had to deal with higher cognitive demands when performing the visual search task. A recent study has shown that while walking, the use of smartphones leads to difficulties in disengaging and reallocating the attentional focus towards the road [83]. Altogether, our results suggest that visual disturbances seem to challenge attentional capacities and affect the ability to maneuver a vehicle in the context of multitasking.

## Impact of the simulated visual acuity higher degradation

To our knowledge, few studies have examined the impact of different levels of visual impairment under high cognitive demands. The impact of visual degradation thresholds has been found on certain tasks related to mobility such as 6/60 or specifying an important visual demand such as reading, i.e. 6/9, face recognition like 6/12. However no single threshold can apply to all types of task to predict performance [84]. Visual acuity below a threshold of 6/9 has been proposed as problematic in terms of driving performance [85]. To add to these observations, in the present study, we found that speed variability was impacted by our manipulation according to the level of visual quality degradation as a targeted by a lower and a higher myopic defocus. This effect was found, however, both during the single and dual task of the highway scenario. The higher visual quality degradation group demonstrated a higher speed variability and a decreased mean speed compared to the lower visual quality degradation group. According to previous research, drivers are able to maintain a stable mean speed

despite very important visual degradation targeting luminance perception in the nighttime but this study did not compare different cognitive load contexts [86]. The results from our study suggest that a strong decrease in visual acuity might increase the attentional resources required for performing multiple tasks while driving. Our results emphasize that the impact of visual load on driving behavior gave rise to dual task interference and that, from a speculative point of view, could eventually induce task prioritization (see Pashler, 1994)—probably in order to efficiently dealing with a limited cognitive capacity [39, 87, 88]. The impact of the lowest visual acuity on speed variability suggests that additional resources are needed for visual integration, but also that resources are allocated to the main task (in this case driving) for security reasons. The cost of visual disruption while driving is an important one to take into consideration for road safety. Individuals driving with non-adapted refractive error corrections exhibit more unsafe driving behavior. For example, adaptation to new lenses to correct refractive errors from astigmatism [89] or myopia have been associated with more unsafe driving behavior and less comfort [61]. However, even though they recognize the importance of clear vision on the road, many drivers report not wearing their optimal correction while driving [90]. Our study emphasizes the importance of education on the potential driving risks associated with poor vision when dealing with the multiple tasks involving high visual and cognitive demands in current vehicles.

## Conclusion

Overall, our results emphasize the importance of meeting visual needs at all distances while driving in demanding visual and cognitive contexts. Our study revealed a detrimental impact of visual quality degradation on driving behavior when young participants were engaged in multiple tasks presented at different visual distances. Indeed, driving behavior was not affected by our visual degradation during a simple driving task involving far vision i.e. in the rural scenario. However, compared to optimal vision, drivers showed larger variability in vehicle maneuvering in response to visual degradation when driving while engaged in a visual search task displayed on a navigation tool at intermediate visual distance i.e. in the highway scenario. This effect remained after controlling the impact of cognitive load. When comparing different levels of visual degradation, the impact of visual load was also observed under greater degradation. The attentional and visual demands of the double task and increased visual demands from our visual degradation manipulation were key elements of our methodology. To our knowledge, this paper is one of the few to demonstrate the emergence of the effects of visual load on driving behavior when controlling for the impact of cognitive load, in the context of a dual task.

The proliferation of peripherals in modern vehicle cabs represents a challenge for drivers in terms of distraction and attention reallocation, and our results suggest that poor visual quality is likely to emphasize these issues, in particular when switching between different tasks and visual distances. Currently, legal visual acuity standards to obtain or renew one's driving license only consider far distance vision. However, new in-vehicle technologies raise the need to consider closer visual demands, such as visual acuity at the intermediate distance and accommodation-convergence capacities. In addition, the numerous difficulties associated with the reduction in visual quality like blur, glare, contrast sensitivity decrease, shortening of the visual field, and the cognitive difficulties associated with aging like managing high cognitive loads, are known to be important challenges among older drivers. We therefore suggest considering intermediate vision needs as well as visual aspects in the development of automobile design as an important human factor to consider in order to improve licensing policies. Moreover, future studies should also examine the impact of physiological visual acuity loss, such as

uncorrected or poorly corrected refractive error in ametropic or presbyopic populations on driving behavior. Our work brings new knowledge to the current optical, ophthalmic and medical literature while highlighting the importance to further investigate road users' visual correction in order to improve their driving safety.

## Acknowledgments

The help and assistance provided by Vadim Sutyushev had been greatly appreciated. We would also like to thank CooperVision for providing support in the form of equipment, precisely the contact lenses used for the degradation of visual quality of participants.

## Author Contributions

**Conceptualization:** Romain Chaumillon, Delphine Bernardin, Jocelyn Faubert.

**Data curation:** Amigale Patoine, Sergio Mejía-Romero.

**Formal analysis:** Amigale Patoine, Laura Mikula, Sergio Mejía-Romero.

**Funding acquisition:** Romain Chaumillon, Jocelyn Faubert.

**Investigation:** Amigale Patoine, Océane Keruzoré.

**Methodology:** Romain Chaumillon, Delphine Bernardin, Jocelyn Faubert.

**Project administration:** Delphine Bernardin.

**Resources:** Romain Chaumillon, Jocelyn Faubert.

**Software:** Sergio Mejía-Romero, Jesse Michaels.

**Supervision:** Laura Mikula, Sergio Mejía-Romero, Jesse Michaels, Delphine Bernardin.

**Validation:** Romain Chaumillon, Delphine Bernardin, Jocelyn Faubert.

**Visualization:** Laura Mikula.

**Writing – original draft:** Amigale Patoine, Laura Mikula, Delphine Bernardin.

**Writing – review & editing:** Amigale Patoine, Laura Mikula, Jesse Michaels, Delphine Bernardin.

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
