## [Decision Letter · Decision Letter 0]

21 Oct 2020

PONE-D-20-29666

Increased visual and cognitive demands emphasize the importance of meeting visual needs at all distances while driving

PLOS ONE

Dear Dr. Patoine,

Thank you for submitting your manuscript to PLOS ONE. After careful consideration, we feel that it has merit but does not fully meet PLOS ONE’s publication criteria as it currently stands. Therefore, we invite you to submit a revised version of the manuscript that addresses the points raised during the review process.

We look forward to receiving your revised manuscript.

Kind regards,

Feng Chen

Academic Editor

PLOS ONE

Journal Requirements:

2) PLOS requires an ORCID iD for the corresponding author in Editorial Manager on papers submitted after December 6th, 2016. Please ensure that you have an ORCID iD and that it is validated in Editorial Manager. To do this, go to ‘Update my Information’ (in the upper left-hand corner of the main menu), and click on the Fetch/Validate link next to the ORCID field. This will take you to the ORCID site and allow you to create a new iD or authenticate a pre-existing iD in Editorial Manager. Please see the following video for instructions on linking an ORCID iD to your Editorial Manager account: https://www.youtube.com/watch?v=_xcclfuvtxQ

3) Please ensure that you refer to Figure 6 in your text as, if accepted, production will need this reference to link the reader to the figure.

4)  Thank you for stating the following in the Acknowledgments Section of your manuscript:

[This work was supported by the Natural Sciences and Engineering Research Council

of Canada (NSERC), NSERC – Essilor Industrial Research Chair (IRCPJ 305729-13),

Research and development cooperatif NSERC – Essilor Grant (CRDPJ 533187 - 2018),

Prompt. LM, JM and AP received support from student awards from the Road Safety

Research Network (Réseau de Recherche en Sécurité Routière) of Québec. We would like

to thank CooperVision who provided the contact lenses for all the participants.]

We note that you have provided funding information that is not currently declared in your Funding Statement. (Equipment from CooperVision).  However, funding information should not appear in the Acknowledgments section or other areas of your manuscript. We will only publish funding information present in the Funding Statement section of the online submission form.

 [This work was supported by the Natural Sciences and Engineering Research Council

of Canada, NSERC – Essilor Industrial Research Chair (IRCPJ 305729-13), Research

and development cooperatif NSERC – Essilor Grant (CRDPJ 533187 - 2018), Prompt

(https://www.nserc-crsng.gc.ca/index_eng.asp & https://www.essilor.ca). The funders

provided support in the form of salary for author DB but had no role in study design data collection and analysis, decision to publish, or preparation of the manuscript.

Authors AP and LM received support from student awards from the Road Safety

Research Network (Réseau de Recherche en Sécurité Routière) of Québec

(https://rrsr.ca/en).]

Additionally, because some of your funding information pertains to commercial funding/equipment supply, we ask you to provide an updated Competing Interests statement, declaring all sources of commercial funding.

In your Competing Interests statement, please confirm that your commercial funding does not alter your adherence to PLOS ONE Editorial policies and criteria by including the following statement: "This does not alter our adherence to PLOS ONE policies on sharing data and materials.” as detailed online in our guide for authors  http://journals.plos.org/plosone/s/competing-interests.  If this statement is not true and your adherence to PLOS policies on sharing data and materials is altered, please explain how.

Please include the updated Competing Interests Statement and Funding Statement in your cover letter. We will change the online submission form on your behalf.

Reviewers' comments:

Reviewer's Responses to Questions

**Comments to the Author**

1. Is the manuscript technically sound, and do the data support the conclusions?

Reviewer #1: Partly

Reviewer #2: Yes

2. Has the statistical analysis been performed appropriately and rigorously? 

Reviewer #1: Yes

Reviewer #2: Yes

3. Have the authors made all data underlying the findings in their manuscript fully available?

Reviewer #1: Yes

Reviewer #2: Yes

4. Is the manuscript presented in an intelligible fashion and written in standard English?

Reviewer #1: Yes

Reviewer #2: Yes

5. Review Comments to the Author

Reviewer #1: The manuscript examines the effects of the interaction between degraded vision and cognitive mechanisms on driving behavior through simulation experiments on people with reduced vision. Research is meaningful, but experimental design requires more contribution. I suggest that the manuscript make major revision to the experimental description.

1. In the experimental design, the 21 participants were selected with varying degrees of vision degradation, no normal vision participation, lack of control group.

2. In the course of the experiment, whether each driver drives all the simulated scenes, continuous driving or random driving.

3. In the study, speed is the characterization parameter of cognitive mechanism, and it is suggested to explain its rationality and comprehensiveness.

4. It is recommended to give a final conclusion on the effects of the interaction between degraded vision and cognitive mechanisms on driving behavior.

Reviewer #2: The topic of this paper is interesting. The methods sound. The results are meaningful and useful. There is one suggestion to improve this paper.

1. The authors need to clarify why only French speakers volunteers or young participants were recruited.

6. PLOS authors have the option to publish the peer review history of their article (what does this mean?). If published, this will include your full peer review and any attached files.

Reviewer #1: No

Reviewer #2: No

---

## [Author Response · Author response to Decision Letter 0]

27 Nov 2020

The response to reviewers was added in a separate document.

---

## [Decision Letter · Decision Letter 1]

17 Dec 2020

PONE-D-20-29666R1

Increased visual and cognitive demands emphasize the importance of meeting visual needs at all distances while driving

PLOS ONE

Dear Dr. Patoine,

Thank you for submitting your manuscript to PLOS ONE. After careful consideration, we feel that it has merit but does not fully meet PLOS ONE’s publication criteria as it currently stands. Therefore, we invite you to submit a revised version of the manuscript that addresses the points raised during the review process.

We look forward to receiving your revised manuscript.

Kind regards,

Feng Chen

Academic Editor

PLOS ONE

Reviewers' comments:

Reviewer's Responses to Questions

**Comments to the Author**

1. If the authors have adequately addressed your comments raised in a previous round of review and you feel that this manuscript is now acceptable for publication, you may indicate that here to bypass the “Comments to the Author” section, enter your conflict of interest statement in the “Confidential to Editor” section, and submit your "Accept" recommendation.

Reviewer #1: (No Response)

Reviewer #2: All comments have been addressed

2. Is the manuscript technically sound, and do the data support the conclusions?

Reviewer #1: Partly

Reviewer #2: Yes

3. Has the statistical analysis been performed appropriately and rigorously? 

Reviewer #1: Yes

Reviewer #2: Yes

4. Have the authors made all data underlying the findings in their manuscript fully available?

Reviewer #1: Yes

Reviewer #2: Yes

5. Is the manuscript presented in an intelligible fashion and written in standard English?

Reviewer #1: Yes

Reviewer #2: Yes

6. Review Comments to the Author

Reviewer #1: 1. In the course of the experiment, whether each driver drives all the simulated scenes, continuous driving or random driving.

2. It is recommended to give a final conclusion on the effects of the interaction between degraded vision and cognitive mechanisms on driving behavior, not just to analyze the data.

Reviewer #2: (No Response)

7. PLOS authors have the option to publish the peer review history of their article (what does this mean?). If published, this will include your full peer review and any attached files.

Reviewer #1: No

Reviewer #2: No

---

## [Author Response · Author response to Decision Letter 1]

25 Jan 2021

The response to the reviewer was added in a separate document.

---

## [Decision Letter · Decision Letter 2]

4 Feb 2021

Increased visual and cognitive demands emphasize the importance of meeting visual needs at all distances while driving

PONE-D-20-29666R2

Dear Dr. Patoine,

We’re pleased to inform you that your manuscript has been judged scientifically suitable for publication and will be formally accepted for publication once it meets all outstanding technical requirements.

Kind regards,

Feng Chen

Academic Editor

PLOS ONE

Additional Editor Comments (optional):

Reviewers' comments:

Reviewer's Responses to Questions

**Comments to the Author**

1. If the authors have adequately addressed your comments raised in a previous round of review and you feel that this manuscript is now acceptable for publication, you may indicate that here to bypass the “Comments to the Author” section, enter your conflict of interest statement in the “Confidential to Editor” section, and submit your "Accept" recommendation.

Reviewer #1: All comments have been addressed

Reviewer #2: All comments have been addressed

2. Is the manuscript technically sound, and do the data support the conclusions?

Reviewer #1: Yes

Reviewer #2: Yes

3. Has the statistical analysis been performed appropriately and rigorously? 

Reviewer #1: Yes

Reviewer #2: Yes

4. Have the authors made all data underlying the findings in their manuscript fully available?

Reviewer #1: Yes

Reviewer #2: Yes

5. Is the manuscript presented in an intelligible fashion and written in standard English?

Reviewer #1: Yes

Reviewer #2: Yes

6. Review Comments to the Author

Reviewer #1: All comments are addressed. I am pleased with the revision. I think this manuscript can be accepted.

Reviewer #2: (No Response)

7. PLOS authors have the option to publish the peer review history of their article (what does this mean?). If published, this will include your full peer review and any attached files.

Reviewer #1: No

Reviewer #2: No

---

## [Editor Report · Acceptance letter]

8 Mar 2021

PONE-D-20-29666R2 

Increased visual and cognitive demands emphasize the importance of meeting visual needs at all distances while driving 

Dear Dr. Patoine:

I'm pleased to inform you that your manuscript has been deemed suitable for publication in PLOS ONE. Congratulations! Your manuscript is now with our production department. 

Kind regards, 

on behalf of

Dr. Feng Chen 

Academic Editor

PLOS ONE